# Understanding the Role of Therapy Dogs in Human Health Promotion

**DOI:** 10.3390/ijerph20105801

**Published:** 2023-05-12

**Authors:** Sonya McDowall, Susan J. Hazel, Mia Cobb, Anne Hamilton-Bruce

**Affiliations:** 1School of Animal and Veterinary Science, Roseworthy Campus, The University of Adelaide, Roseworthy, SA 5371, Australia; susan.hazel@adelaide.edu.au; 2Animal Welfare Science Centre, Faculty of Veterinary and Agricultural Science, The University of Melbourne, Parkville, VIC 3010, Australia; mia.cobb@unimelb.edu.au; 3Adelaide Medical School, The University of Adelaide, Adelaide, SA 5005, Australia; anne.hamilton-bruce@adelaide.edu.au

**Keywords:** animal-assisted therapy, animal-assisted interventions, regulations, legislation

## Abstract

Dogs may provide humans with a range of physical, mental and social benefits. Whilst there is growing scientific evidence of benefits to humans, there has been less focus on the impact to canine health, welfare and ethical considerations for the dogs. The importance of animal welfare is increasingly acknowledged, indicating that the Ottawa Charter should be extended to include the welfare of non-human animals supporting the promotion of human health. Therapy dog programmes are delivered across a variety of settings including hospitals, aged care facilities and mental health services, highlighting the important role they play in human health outcomes. Research has shown that that there are biomarkers for stress in humans and other animals engaged in human–animal interactions. This review aims to assess the impact of human–animal interactions on therapy dogs engaged in providing support to human health. While challenging, it is paramount to ensure that, within the framework of One Welfare, the welfare of therapy dogs is included, as it is a key factor for future sustainability. We identified a range of concerns due to the lack of guidelines and standards to protect the wellbeing of the dogs engaged in these programmes. Extension of the Ottawa Charter to include the welfare of non-human animals with leveraging through a One Welfare approach would promote animal and human health beyond current boundaries.

## 1. Introduction

Within Australia, it is estimated that 40% of households have a dog [1], and dogs may provide humans with a range of physiological, psychological and social benefits [2,3,4,5]. While there is growing scientific evidence of these benefits to humans, the health and welfare impacts on dogs and ethics of engaging dogs in these ways have been less explored [6,7,8]. Globally, there is a rapidly expanding field whereby dogs are being engaged to support humans in various therapy roles [9]. Currently, in Australia, therapy dog programmes are mostly unregulated. This means that uniform minimum standards or codes of practice to ensure the safety and welfare of the dog are not present [9]. As there are no existing registration or accreditation requirements, identifying and understanding the current population of working therapy dogs within Australia is challenging. 

The importance of a One Welfare [10] approach is increasingly acknowledged, indicating a need for the Ottawa Charter [11] to be extended from a human health focus to include non-human animal welfare when supporting the promotion of human health. It takes into account the physical, mental, and emotional health of humans and animals, as well as the impact from, and of our actions on, the environment [12]. One Welfare [10] recognises that our interactions with animals affect them as individuals, that different species interact with each other and that humans are able to benefit from these interactions [13]. One Welfare is increasingly being acknowledged [12], indicating the need for the Ottawa Charter [11] to be extended to include non-human animal welfare with respect to supporting the promotion of human health [10]. The importance of understanding animal welfare and the potential impact of physical and mental harm when working in therapy roles with people is paramount to ensure the wellbeing of non-human animals for that work with humans to be sustainable [7].

Therapy dog programmes are delivered across a variety of settings including hospitals, aged care facilities, schools and tertiary education centres, mental health services, courthouses and airports [14]. Outcomes that have been reported by people who are involved in a therapy dog programme include: enhanced socialisation, improved mood, decreased symptoms of depression and/or anxiety, increased independence, enhanced self-esteem and decreased feelings of anger [15,16,17]. Definitions of therapy dogs and their roles vary throughout the world [18]. Recently, over 70 global experts in the field of anthrozoology published definitions to clarify the terms and associated roles to limit confusion and clarify definitions for animals who support people [19]. Given the historic variety of terminology, the following definition of a therapy dog will be used for this review, based on Howell, Nieforth [19]: ‘*an animal who is included into the work of a qualified health professional in the provision of a structured, goal-directed treatment*’.

We undertook a literature review to assess the impact of human engagement on therapy dogs enrolled in providing animal therapy support to human health. This is paramount to ensure that, within the framework of One Welfare [12], the welfare of therapy animals is included, as it is a key factor for future sustainability of such programmes [20]. Major literature databases, including but not limited to Cochrane, Scopus, Science Direct, PubMEd and Google Scholar, were searched online. Search keywords included: “animal-assisted therapy”, “animal-assisted interventions”, “regulations”, “legislation”, “therapy dog”, “therapy canine”, “cortisol”, and strategies such as Boolean and snowball searching were used. We address our findings by focusing on therapy dog issues related to welfare and ethics, stress, and management of risks, health, and safety. All of these highlight the need for mechanisms such as an extension of the Ottawa Charter [11] to address potential issues by enabling an industry-engaged code of conduct to be developed to provide a formalised One Welfare [12] approach to promote health for all. This paper proposes to further link One Welfare and Health Promotion to strengthen the development of animal welfare policies and practices that concurrently prioritise the wellbeing of both humans and animals and promote their coexistence and vitality [21,22].

## 2. Why Are Welfare and Ethics Important to Therapy Dogs?

The welfare and ethical engagement of animals for health outcomes and benefits to humans, including therapy dog programmes, is a topic of community interest [23]. The achievement of animal welfare protection is based on the five domains of animal welfare: nutrition, physical environment, health, behavioural interactions, and mental state [24]. These domains serve as a structured and systematic means of assessing animal welfare, considering both the positive and negative experiences that animals may encounter, which are known as their affective states [24]. Additionally, these domains consider the impact that humans have on these states [24]. This expands on animal welfare as being understood as both a scientific concept and the lived experience of an animal. It is influenced by value-based ideas (health and functioning, agency in behavioral interactions with people, other animals and their environment, affective states and natural living) which are considered important elements for an animal to experience a good life [25], where they experience more positive feelings than negative ones. Measurement of welfare in therapy dogs needs to be based on indicators arising from robust animal welfare science that demonstrate not only an absence of pain and suffering, but also evidence of a positive and enriched life [26,27]. 

The ethical challenge posed by dogs’ participation in therapy roles links to the tension of humans engaging with animals to achieve their own goals without the dog having autonomy to choose or reject its role [28]. The field of therapy dogs lacks guidance presented in the ethical framework of other human–animal relationships [29]. In particular, the ethical issues that arise in therapy dog programs is the potential exploitation of the dog as a therapeutic tool [29]. There is a risk that dogs may be viewed solely as tools to provide emotional or therapeutic support rather than as individuals with their own needs and desires. This could lead to the neglect of the dog’s welfare, such as ignoring signs of stress or overworking the dog. This then highlights the importance of daily interactions and demands, such as feeding, grooming and travel, that must be appropriate and reasonable to prevent injury or illness to the dog [7,23,28]. A recent study highlighted the wide discrepancies in guidelines and standards held by organisations relating to therapy dogs in Animal Assisted Interventions (AAI) in the United States [30]. These findings were consistent with four key factors which have been raised throughout the literature [6,7,23,30,31,32,33], including dog health and safety, welfare and ethics, handler training and education, and dog behavioural selection. There is a clear need to ensure that the integration of therapy dogs into therapy programmes upholds the highest degree of dog welfare [34]. This is of particular importance given societal expectations and community attitudes towards animals [20]. The welfare of therapy dogs has been identified as a key factor for the future sustainability of these practices due to the connection between public expectations and social license to operate [20,35].

People’s perception of dog welfare varies based on the role and context of the dog [20]. This may reflect an assessment of the perceived intrinsic and extrinsic significance of the dog’s role in society [36]. Canine welfare-related concerns in therapy programmes across the USA which have been identified were limited access to water, high temperatures in environments such as nursing homes, and extended periods for which dogs were involved in therapy sessions [37]. Across the existing evidence base, concern for dog welfare has traditionally focused on aversive and coercive dog handling techniques, such as the use of choke chains, shock collars, loud reprimand, physical correction along with dogs being unable to avoid social intrusions or seek refuge [7,23,38,39]. Our current knowledge of the impact of therapy programmes on the welfare of dogs is limited [40]. This may be due to funding prioritising the focus on human benefits from such programmes, or a lack of validated measures that are readily available to determine the canine welfare experience [41]. Although a dog may not show signs of aggression during a therapy programme, this does not mean that the animal is enjoying it [40]. 

## 3. What Is a Dog’s Stress Response and How Is It Measured?

When humans are faced with a stressful or unknown situation, the Hypothalamic–Pituitary Adrenal (HPA) axis is activated and the steroid structured hormone cortisol is released [42,43]. Like in humans, stress responses in dogs result in cortisol release from the HPA axis [44]. Cortisol secretion may function as a coping mechanism in response to an endogenous or exogenous threat [15]. This can result in elevated cortisol in acute stress situations and reduced or dysregulated cortisol production over periods of chronic stress as the HPA axis becomes exhausted, limiting the value of cortisol when used as a sole measure of stress in dogs and other mammals [45]. Although stress as a state of arousal can be considered positive or negative, sustained physiological stress can result in long-term health issues, such as diseases of the gastrointestinal, cardiovascular, urinary and immune systems [6,15]. The role of assisting people as a therapy dog may result in the dog experiencing detrimental stress that negatively impacts its welfare [15]. Interestingly, studies in humans investigating stress biomarkers found that when people are engaged with interacting or stroking a dog, sensory stimulation in both the human and the dog can activate oxytocin and decrease cortisol levels in humans; however, it is important to consider the role of other physiological systems [46,47,48,49], although an increase has been observed at times in dogs [48,50]. However, the experience of dogs involved in a therapy dog programme is dependent on the individual dog’s genetics, its training and previous experiences, the handler, the environment, the type of therapy interaction, tasks undertaken, duration, and also the patient when in a clinical therapy setting [40].

The main indicators of the stress response typically measured in therapy dogs are physiological (e.g., salivary cortisol, heart rate and behavioral (e.g., observations of validated stress-related behaviours)) and/or indicated in handler reports, as shown in Table 1. Salivary cortisol is a widely used indicator in dog research as reflected in Table 1, but varies significantly based on the individual dog, as well as the experimental and environmental conditions [45]. Additionally, since cortisol levels can respond to both negative and positive stimulation, it is problematic to utilise it as a standalone indicator [51]. However, there is a need for caution when interpreting salivary cortisol as a quantifiable measure of welfare in isolation as its levels could be the reflection of a variety of reasons [6,45]. Best practice for assessing animal welfare is a combination of multiple physiological measures and behavioural observations [45,52]. This is rarely reported in the assessment of therapy dog programmes.

Heart rate is a psychophysiological measure of affective and cognitive responses in animals [52], but its use as a valid indicator to measure the stress response in dogs is varied, with studies identifying little correlation to stress response [60,61]. No correlation between stress-related behaviours (licking lips, yawning, grooming and withdrawal) and heart rate was reported; however, only one dog participated in this study, highlighting a possible study bias based on the behavioural traits of the dog involved [52]. In contrast, one study identified an increase in therapy dogs’ heart rate on therapy days compared to control days based on observations of four dogs [59].

It is also difficult to compare results from the published studies of welfare indicators in dogs during therapy due to variations in methodology. For example, the duration of the therapy in reported studies varied from 20 min to 8 h, along with differing intervals in session frequency varying from daily, weekly, biweekly, to sessions over an extended period [15,44,51,53,54,55,56,57,58,59]. Furthermore, the behavioral observations and assessment to identify suitable dogs also varies across studies as there is no specific behavioural assessment designed for recruitment into therapy programmes [51].

## 4. Factors Affecting Therapy Dogs’ Stress Responses

### 4.1. Handler

The importance of handlers has been highlighted in monitoring and assessing their dog’s behaviour. This enables early intervention to help prevent stress responses that may negatively impact the dog’s welfare [40,52,57,62]. One study has identified the potential impact of a handler’s sex on the dog’s salivary cortisol concentrations [63]. In this study, a female handler–male dog combination was observed to produce lower salivary cortisol variability [63]. The effect of handler sex has also been identified in rodents, with female handlers effecting a lower stress response in the animals than male ones [64]. Studies have also noted that dogs displayed sensitivities and stress as a reflection of their owner’s wellbeing, which speaks to the role of attachment between dog and handler and could also suggest emotional contagion between the dog and its owner [27,62,63]. 

Handlers with higher levels of education were more aware of stress responses in dogs than those with lower education levels [6]. Two studies also noted the Dunning–Kruger effect, in that handlers without the required knowledge or training may not be able to identify fear or stress reactions, or even unwittingly exacerbate them, and thus have a detrimental effect on the dog’s wellbeing [7,51]. The importance of continuing education and certification in this field is deemed significant [65] and should be mandatory.

### 4.2. Recruitment and Selection

Seven standard behavioural dimensions have been outlined as a general test for therapy dogs: reactivity, fearfulness, activity, sociability, responsiveness to training, submissiveness, and problem behaviours [66]. Whilst dogs are assessed on their behavioural responses in a range of different environments, one study argues that this does not guarantee that the dog will not become stressed [44]. Furthermore, it has been argued that a test used to assess a dog’s suitability for therapy programmes should be based on how suitable and capable the dog is in performing the tasks associated to a particular role [67]. 

Canine personality is an underexplored area in the field of successful therapy dog work. Personality is a more stable indicator of reaction to show different behavioural responses to a stimulus; while certain behavioral dimensions are used to assess the suitability of a dog for therapy work, the variability or uniformity of personalities among dogs that excel in therapy work has not been extensively studied [68]. Understanding the personality traits that are most beneficial for therapy work could aid in the selection and training of therapy dogs [51,68,69]. It is possible that certain personality traits, such as high sociability and low fearfulness, are more conducive to successful therapy work [6]. Further research is needed to determine whether there are specific personality traits that are most important for successful therapy work and to learn to identify these traits during the selection process.

In addition to personality traits, the attachment between the handler and the therapy dog is also an essential factor that can influence the dog’s effectiveness [6,70]. A therapy dog is not simply a tool that can be taken off the shelf and deployed equally in all situations [70]. Like other working dogs, such as military and police dogs, therapy dogs may work differently depending on the people they work with [31]. The handler’s behavior, tone of voice, and body language can all impact the dog’s behaviour and response. A strong bond between the handler and dog can enhance the effectiveness of the therapy dog and improve the overall experience for the individuals they are working with [70].

Therefore, it is crucial to not only assess the personality traits of a therapy dog but also consider the ability of the handler along with the attachment between the handler and the dog during the selection process. An assessment of the handler’s ability to read the dog’s behaviour and respond appropriately is essential in ensuring the dog’s wellbeing and effectiveness as a therapy dog [70].

### 4.3. Organisational Perspective

Each organisation has its own perspective and criteria for appropriate dog behaviour and the identification of canine stress, creating challenges such as organisational bias and limited independent verification [70,71]. Larger organisations utilising national recruitment programmes identified the same challenges, whereas smaller organisations use their own (often not evidence-based or validated) independent criteria [14]. This is in contrast to a more recent study in the United States, in which most organisations, regardless of size, were compliant with current guidelines requiring formal evaluation of dog behaviour and temperament [30]. The adoption of common guidelines or protocols for recruitment of dogs engaged in therapy programmes is paramount to ensure that welfare outcomes for the dog are achieved [14,70,71]. 

## 5. Managing Risks, Health, and Safety

There are multiple issues raised in relation to the health risks which dogs present in therapy dog programmes. Possible risks to humans include transmission of zoonoses, participant allergies, fleas, scratches and bites, but there has been little focus on the risk to the health and safety of the dog [7]. Zoonotic infection can transmit both ways (human to animal, animal to human), and it is possible for both human and non-human animal participants within therapy dog programmes to be at risk [72]. Whilst it has been identified that therapy dogs could act as reservoirs or carriers of infectious agents, the impact of infections such as multidrug-resistant bacterial infections in animals is just as important, particularly when deployed in environments such as hospitals [73]. Although rates of infection did not change during therapy programmes, there is a lack of reporting and validity regarding the surveillance methods used [73].

Feeding raw meat diets and the potential for subsequent zoonotic transmission is also an important consideration. Although controversial, raw meat diets can be perceived as nutritionally superior and providing health benefits to dogs [74,75,76]. Participants in contact with therapy dogs may be immunocompromised, thus the shedding of pathogens by dogs poses an increased health risk [74,77]. Between 21% and 44% of raw meat diets are contaminated with *Salmonella* spp., posing a risk to patients [74], and it has been identified that *Clostridium*, *Giardia*, and *Salmonella* could also be isolated from therapy dogs (*n* = 80/102) [78]. In an earlier study, dogs fed a raw meat diet experimentally contaminated with *Salmonella* spp. were 11.4 times more likely to shed this pathogen within a week of consumption [79]. Understanding the risks involved appears to be limited as only 13% of smaller therapy organisations in the USA (*n* = 24) have a policy that prohibits the feeding of raw meat diets [30].

The most common health-related protocols reported for therapy dog programmes is the requirement for regular veterinary visits and vaccinations [80,81]. However, programmes exist which require several health assessments involving faecal and nasal cultures, vaccinations including canine distemper, parainfluenza, parvovirus, hepatitis and rabies, preventative treatments include flea and tick control and heartworm [78]. There is variability in practice with only 63% of organisations requiring vaccinations, 75% requiring negative faecal parasite result, 54% requiring flea and tick preventative treatments, and 42% requiring the dog is bathed prior to the visit [30]. 

## 6. Future Directions for Research

Evidence-based standards of practice to ensure minimum protections for animals engaged in animal-assisted interventions/programmes including animal-assisted therapy have been developed internationally [82]. However, within Australia, at present, there are no regulations or legal requirements to protect animals engaged within animal-assisted interventions/programmes, which highlights an important area for future investigation and intervention.

The Ottawa charter could be modified to advocate for animal-friendly policies to enable the integration of animal-assisted therapy programs within the health setting in the acute and community setting. Should this be achieved, it would be paramount to ensure that the welfare standards of the animals involved to support health promotion outcomes are a key outcome in safeguarding the animal.

## 7. Conclusions

The Ottawa Charter commits to health promotion through the promotion of health and the collaboration and partnership among all sectors of society, through creating supportive social and physical environments to enable people to achieve their full potential for health. Therapy dog programs can benefit a diverse group of individuals who may need emotional support, companionship, and assistance in managing their health and wellbeing. Therapy dogs offer a unique form of support that can complement traditional medical treatments and provide individuals with increased control over their health and environment, enabling them to make healthier choices. Per the Ottawa Charter, responsibility for health promotion is shared between individuals, community groups, health professionals, health service institutions and governments who must work together towards a health-care system that contributes to the pursuit of health. The important role that therapy dogs play to support and enhance the lives of humans has been well documented. This has led to an increase in popularity of dogs engaged in different types of programmes, in various environments, for a variety of support needs. A range of concerns due to the lack of guidelines, certification and standards to protect the welfare of dogs engaged in these programmes has been identified. Ignoring these concerns and the risk to the dogs involved may lead to inappropriate engagement of dogs within programmes, increased negative behavioural responses from dogs and possible harm to both dogs and humans. To ensure we can meet societal expectations and secure the future sustainability of therapy dog programmes for those in need of such therapy, transparency and assurance of good animal welfare is a key priority. Extension of the Ottawa Charter to include non-human animals and their welfare, with leveraging through a One Welfare approach, would promote human health beyond its current boundaries.

## Figures and Tables

**Table 1 ijerph-20-05801-t001:** Physiological stress studies investigating salivary cortisol in dogs during therapy programmes (chronological order).

Author	Study Design	n	Duration	Welfare Indicators	Key Findings
Haubenhofer and Kirchengast [53]	Cross-sectional (Any breed, age, sex)	18	1–8 h	Salivary Cortisol	Increased salivary cortisol on therapy days vs. control days Higher salivary cortisol in short sessions compared to longer sessions
Haubenhofer and Kirchengast [54]	Cross-sectional (Any breed, age, sex)	18	1–3 h	Salivary Cortisol Handler emotion questionnaire	Increased salivary cortisol on therapy days vs. control days Increase in salivary cortisol if high frequency of sessions
King, Watters [15]	Cross-sectional (Any breed, age, sex)	21	2 h	Salivary Cortisol Behaviour Handler behaviour evaluation survey.	Increase in salivary cortisol after 60 min No effect from a time-out session Dogs > 6 years of age showed increased behavioural signs associated with stress
Glenk, Kothgassner [55]	Cross-sectional (Any breed, age, sex)	21	50–60 min	Salivary Cortisol	No difference in cortisol levels between days worked and days not worked Decrease in cortisol level when off lead
Glenk, Kothgassner [56]	Cross-sectional (Any breed, age, sex)	5	55–60 min	Salivary Cortisol Behaviour	No changes in behaviour Decrease in salivary cortisol levels in the last two sessions
Ng, Pierce [57]	Cross-sectional (Any breed, age, sex)	15	30–60 min	Salivary Cortisol Behaviour	Increase in salivary cortisol within the novel setting No difference between working vs. non-working day
Koda, Watanabe [58]	Cross-sectional (Any breed, age, sex)	47	70 min	Salivary Cortisol Handler questionnaire	Decrease in salivary cortisol following sessions Link identified between dog stress levels and handler self-reported stress levels
Pirrone, Ripamonti [59]	Cross-sectional (Any breed, age, sex)	4	55 min	Salivary cortisol Behaviour Assessment Heart Rate	Increased heart rate on therapy days vs. control days Increase in salivary cortisol before therapy intervention not statistically significant (*p* > 0.05)
McCullough, Jenkins [51]	Cross-sectional (Any breed, age, sex)	24	Various time points used.	Salivary Cortisol Handler self-report questionnaire Behaviour Assessment and research questionnaire (C-BARQ) Behavioural Ethogram	No change in cortisol over time in study Female dogs with significantly lower cortisol levels than males Lower cortisol levels in older dogs Lower cortisol levels associated with increased affiliative behaviour Higher cortisol levels with stress behaviors
Clark, Smidt [44]	4 × 4 Latin Square (Any breed, age, sex)	4	15 min	Salivary Cortisol	Decrease in salivary cortisol when visiting same environment twice a week

## Data Availability

Not applicable.

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
