# Peer review of "Understanding the Role of Therapy Dogs in Human Health Promotion"

_ijerph, 2023, doi:10.3390/ijerph20105801_

Round 1

Reviewer 1 Report

Thank you for this interesting review on a very important issue. There have been previous reviewes in the area (for example Glenk, 2017, Glenk and Foltin, 2021, and others), but it is worth publishing this in different Journals and from different perspectives.

There is some major remarks and some minor remarks on this review.

I miss a section on how you can prevent low welfare in the therapy dogs. Making risk assessments before you start an intervention is mandatory for many interventions that I come across. There are developed risk assessment protocols which have to be signed by the dog handler and the responsible person at the place where the intervention will take place.

I also wonder how you look upon having standards for testing dogs for suitability before allowed to start an education and standards for content and length of special educations. We are struggling in my country with a lack of regulations in this area, as you also mention in your review. 

Minor comments:

Line 26: I would delete ", has been identified", as you write this in the beginning of the sentence.

Line 52-55: This sentence seems a bit repetitive of the sentence on line 44-47.

Line 244: Add space between "with" and "(44)".

Line 259: Add space between "(32)" and "in the".

Line 266: I think there is missing a word at the end of this sentence. Maybe adding "meet" or "experience" would solve the problem.

Line 331-482: I read through the reference list and found that you have been quite inconsistant in how you write titels and journal names. Usually Journals want authors to write titels with only small syllables and the Journal name with large syllables in the beginning of each word except for words like "of".

Line 406: Do you need to have the shortening of the Journal name here?

Line 476: Do you need to write the French name of the Journal?

The review is very well written with a good English.

Reviewer 2 Report

Regarding the conflict of interest issue, I am the founder of a nonprofit the mission of which is to advance and share knowledge that improves the lives of animals.  

This is a welcome literature review addressing the need for additional studies that provide evidence regarding the welfare of dogs involved in AAT. 

The review is clear and thorough. 

Author Response

We would like to thank the reviewer for their appreciation of our work. It is very much appreciated.

Reviewer 3 Report

This is a very important paper.  I have a few very minor suggestions, and two more substantial ones.

-Please review the abstract for clarity of wording.
-Please review line 37 for clarity of wording?
-Does line 88/89 need an and?
-Line 169/190/etc – reword or just the way source authors?
-The attention to the handler, indeed, should be assessed too!  Line 247-251.
-Line 92/125 – USE therapy dogs for human benefit? Is this language not counter to what is being argued.  Using the word use? Consider using wording such as "integrated" into social interactions, therapy sessions, etc.
-Can you give a bit more background on how this is a narrative/literature review.  I f it is a narrative review, whey is it also referred to as a literature review. 
-Line 231 to 239 – I am confused by the behavior assessment being separated from personality assessment of the dog.  Are the tests not looking for both?  Maybe it is different in the UK.  This is not a service dog task oriented test, personality should always be a part of the test, given the dogs' interaction with people.  Our test, although focused on behavior, is a personality test of sorts – main question being, is the dog having fun too?

-Line 70 and 71 - this is a very narrow definition of a therapy dog, defining it as AAT, which is goal directed and with a professional (not a para-professional). I am not convinced that the literature reviewed is only specific to AAT, and not AAA (animal assisted activities, or therapy dog visiting programs). This needs to be clarified. 
-Do you have any suggestions on how the charter should be expanded/modified?
